# Markerless Laryngeal Motion Tracking During Swallowing Using an RGB-D Camera with 3D Head-and-Neck Alignment

Tomoya Onishi
*Graduate School of Information*
*Science and Technology*
*The University of Osaka*
Osaka, Japan
onishi.tomoya@ist.osaka-u.ac.jp

Yoshihiro Midoh
*Graduate School of Information*
*Science and Technology*
*The University of Osaka*
Osaka, Japan
midoh@ist.osaka-u.ac.jp

Jun Shiomi
*Graduate School of Information*
*Science and Technology*
*The University of Osaka*
Osaka, Japan
shiomi-jun@ist.osaka-u.ac.jp

Toru Yuba
*Professor emeritus*
*Mie University*
Hyogo, Japan
t-yuba@yubamethod.org

Noriyuki Miura
*Graduate School of Information*
*Science and Technology*
*The University of Osaka*
Osaka, Japan
nmiura@ist.osaka-u.ac.jp

*Abstract*— **Dysphagia, or difficulty in swallowing, can lead to aspiration pneumonia—a lung infection caused by inhalation of food or fluids— particularly in older adults, making its early detection and prevention critically important. While invasive techniques such as X-ray fluoroscopy are traditionally used to evaluate swallowing function, there is a growing need for simple and noninvasive methods are needed not only for screening but also for monitoring the effectiveness of swallowing rehabilitation exercises such as effortful swallowing or vocal training. In this study, we propose a markerless laryngeal motion tracking method using a noncontact RGB-D camera combined with 3D head–neck model alignment. A subject-specific 3D model is constructed from depth images acquired in a static state, and swallowing-related laryngeal motion is extracted by compensating for forward–backward body sway and vertical neck movement through virtual model manipulation. In experiments using a water drinking test, the proposed method effectively reduced motion artifacts from body and neck movement and enabled clearer extraction of laryngeal motion associated with swallowing. A comparison across three subjects revealed consistent trends in skin displacement patterns corresponding to muscle activity during swallowing.**

*Keywords— aspiration, swallowing function test, depth sensor, markerless motion tracking, virtual 3D model*

## I. INTRODUCTION

Maintaining the ability to eat independently throughout life is a cornerstone of healthy aging, critically impacting quality of life and purpose in life. In high-income countries, aspiration pneumonia (pneumonia caused by accidental ingestion of food that should have entered the stomach through the esophagus into the lungs) accounts for most pneumonia-related deaths among older adults. Globally, 2.18 million people died of lower-respiratory infections in 2021 [1], and clinical estimates indicate that 5–15 % of pneumonia cases are aspiration-related, translating to roughly 110,000–220,000 deaths each year [2]. In the United States, 1.11 million aspiration-pneumonia deaths were registered from 1999 to 2017 ($\approx$ 58,600 annually), with only a modest decline in age-adjusted mortality rates [3]. Japan reported more than 40,000 aspiration-pneumonia deaths in 2020 [4]. In Europe,

131,450 people died from pneumonia in 2016 [5], this corresponds to roughly 7,000–20,000 aspiration-pneumonia deaths annually. With increasing longevity and aging populations, aspiration is becoming an increasingly common global health concern.

Because dysphagia often progresses without noticeable symptoms [6], continuous real-world monitoring and early intervention are essential for preventing aspiration pneumonia. video fluoroscopic swallowing studies (VFSS) involve the ingestion of a contrast-enhanced meal under X-ray fluoroscopy, during which the swallowing process is recorded and evaluated. However, VFSS requires hospital visits and exposes patients to radiation. In recent years, less invasive devices—such as those using ultrasound or wearable accelerometers—have been developed. Nevertheless, most of these still require skin attachment and skilled operation for measurement and analysis, limiting their applicability in home settings.

To address this challenge, we propose a markerless method for tracking laryngeal motion during swallowing, using an RGB-D camera combined with 3D head–neck model alignment. While markers are typically employed to correct for body motion [7], our approach constructs a personalized 3D model of the head and neck from camera images and isolates laryngeal movement by virtually reconstructing and subtracting the forward–backward and vertical movements of the head and neck. This noncontact, low-cost method enables head motion compensation with millimeter-level accuracy, making it a promising platform for home-based rehabilitation and large-scale screening to mitigate the global burden of aspiration pneumonia.

In Section 2, we summarize previous studies related to measuring laryngeal movement. In Section 3, we explain the details of the proposed method. In Section 4, we report the experimental results, and finally, we summarize our findings in Section 5.

## II. RELATED WORK

In accordance with established clinical guidelines [8], the diagnostic process for dysphagia in the hospital setting typically begins with a clinical interview to assess

This material is based upon work supported by XING INC.

swallowing-related symptoms. If no signs of dysphagia are suspected, the patient is placed under observation. If dysphagia is suspected, further assessments are conducted, including evaluations of cognitive and physical function, as well as examinations of the oral cavity, pharynx, and larynx. The functional assessments focus on identifying contributing factors beyond the primary disease, such as cognitive impairment and postural instability. Examination of the swallowing structures is typically performed using endoscopy. When patients are unable to undergo VFSS or flexible endoscopic evaluation of swallowing (FEES), simplified bedside tests—such as water or food swallowing trials—are employed. If VFSS or FEES is feasible, these tests are performed to obtain detailed information for planning appropriate clinical interventions.

In the hospital setting, invasive but highly accurate tests are used for definitive diagnosis, whereas noninvasive methods with lower patient burden are employed for preliminary or screening purposes. Our objective is the routine monitoring of swallowing function, for which we adopt a noninvasive approach. A summary of existing minimally invasive devices and previously proposed methods is provided in Table 1. H. Hashimoto et al. proposed a method for simultaneously measuring mouth and laryngeal movements during swallowing [7]. Their approach involved attaching an ECG sensor and a microphone to the neck to measure vocal cord vibrations and swallowing sounds, respectively, while tracking laryngeal motion using a Kinect sensor and three markers affixed to the neck. Aihara et al. developed a device consisting of five vertically aligned stretchable strain sensors placed in contact with the laryngeal region to measure strain associated with laryngeal elevation [9]. N. Kuramoto et al. estimated swallowing duration by combining a neck-mounted microphone with deep learning-based analysis of swallowing sounds [10]. In addition, methods using surface electromyography (sEMG) electrodes placed on the skin have been proposed to record the activity of muscles involved in swallowing [11,12]. Omori et al. utilized vertically arranged photoelectric distance sensors to detect midline changes in the neck during swallowing [13]. The four supporting legs of their device maintain fixed contact with the neck, enabling distance-based observation of laryngeal motion.

The aforementioned methods enable the measurement of bone and muscle movements associated with swallowing using various types of sensors. However, many of these require direct contact with the body or subject immobilization during measurement. Even partial contact between sensors and the skin can interfere with the intake of liquids or food and training. Fig. 1 presents representative training methods for enhancing swallowing function and preventing aspiration. The Forehead and Shaker exercises aim to maintain neck muscle load by pressing down on the head or lifting it, respectively. Voice training is conducted through singing, while the Mendelsohn maneuver involves sustaining laryngeal elevation to strengthen the muscles involved in swallowing. Since these exercises often require substantial body movement, minimizing physical contact and restraint is highly desirable. To address this, we propose a fully contactless method that uses only a camera to capture swallowing-related motion by compensating for body movements, eliminating the need for physical attachment or fixation.

## III. METHOD

The skin surface lacks distinct landmarks, such as corners with sharp brightness gradients, which are commonly used in markerless motion tracking. To address this limitation, we constructed a 3D model of the head and neck that incorporates anatomical structures, including muscles and nerves involved in swallowing, as illustrated in Fig. 2. The model was based on a publicly available male head model [14] and it was developed using Blender, with the insertion of pseudo-bones to allow for the representation of postural fluctuations and vertical neck movements. The 3D model is adjusted in virtual space to align with the depth information of the laryngeal skin surface obtained from the RGB-D camera.

An overview of the proposed method is presented in Fig. 3. The Intel RealSense D405 is employed as a short-range RGB-D camera. We have reported that the error in measuring laryngeal motion using VFSS with the same camera ranged from 0.88 to 1.82 mm, indicating millimeter-level accuracy [15]. In this work, the 3D model of the head and neck is

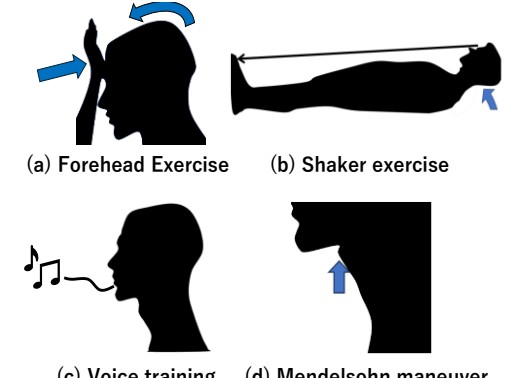

(a) Forehead Exercise  (b) Shaker exercise

(c) Voice training  (d) Mendelsohn maneuver

Fig. 1.  Typical swallowing function training methods for preventing aspiration.

TABLE I.  MINIMALLY INVASIVE LARYNGEAL MOVEMENT MEASUREMENT METHODS

| Method | Contact | Restraint | Principle of operation |
|---|---|---|---|
| [7] | Yes | No | Multimodal (Kinect, Electroglottogram, Microphone) |
| [9] | Yes | Yes | Elastic strain sensor |
| [10] | Yes | No | Contact Microphone Postural gyro sensor |
| [11] | Yes | Yes | Electromyogram |
| [12] | Yes | Yes | Electromyogram |
| [13] | Yes | Yes | Photoelectric distance sensor |

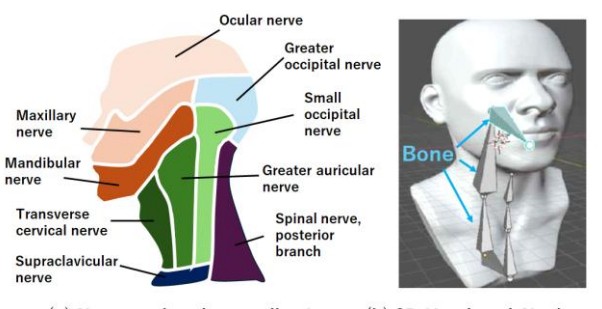

(a) Nerves related to swallowing  (b) 3D Head-and-Neck model

Fig. 2.  3D Head-and-Neck model with adjustable joint orientation.

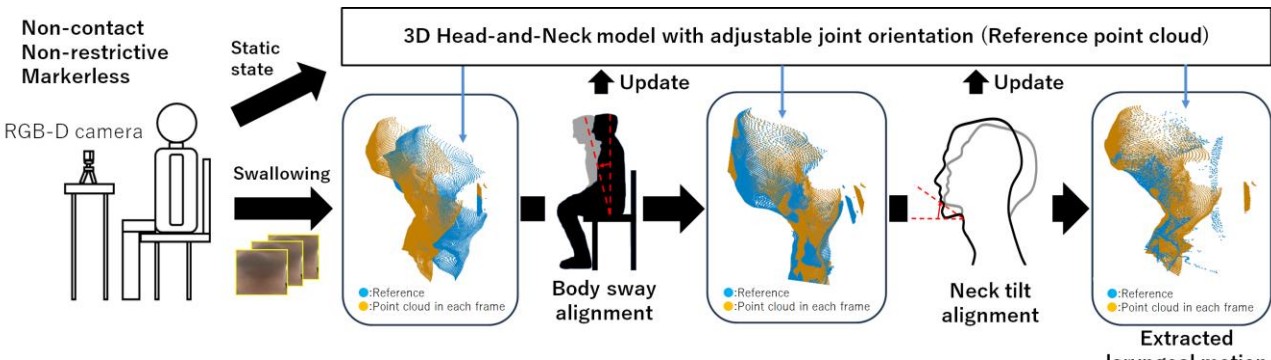

Fig. 3. Alignment of 3D models and extraction of laryngeal movements using an RGB-D camera based on the proposed method.

adjusted by comparing it with the acquired depth map, and fluctuations in the laryngeal region are extracted relative to a reference static state. In the following sections, we first describe the construction of the 3D head and neck model, and then explain the procedure for compensating for body sway and neck movement.

### A. 3D Head-and-neck Model Construction from a Static-state Depth Map

To improve the accuracy of positional correction, it is essential to account for the subject's skeletal structure, head orientation at the time of imaging, and the spatial relationship between the camera and the subject. Accordingly, the scale, orientation, and initial head position of the 3D model are calibrated based on a reference neck mesh. Fig. 4 illustrates a subject-specific 3D model in a static (not swallowing) state after correction. In the figure, the blue mesh represents the reference depth map.

### B. Body Sway Alignment

Postural fluctuations are corrected using the iterative closest point (ICP) algorithm [16]. The ICP algorithm aligns two point clouds by iteratively performing two steps: (1) identifying the closest corresponding points between the source and target point clouds, and (2) minimizing the distance between these matched pairs through spatial transformation. This process enables the removal of translational and rotational differences. In our method, ICP is applied to each point cloud representing laryngeal movement frames, aligning them to a static reference point cloud.

### C. Neck Tilt Alignment

This section describes the correction of neck tilt using a 3D model. First, the neck coordinates of the 3D model are aligned with a reference coordinate system, as illustrated in Fig. 4. Then, the model's bones are rotated to minimize the distance between the point with the smallest z-coordinate in each frame (corresponding to the lower jaw) and the point with the smallest z-coordinate in the static 3D reference model. The rotation is performed incrementally (gradual adjustment of the neck bone angle until the difference between the camera surface and 3D model surface falls within 5mm), and is terminated when the z-axis difference between the two points falls within 5 mm. The neck coordinates after bone adjustment are regarded as the corrected coordinates. This correction enables the transformed point cloud to be converted into the reference coordinate system, compensating for neck movement.

### D. Laryngeal Motion Extraction

Laryngeal movement during swallowing is extracted by subtracting the reference depth map—corrected through the procedures described above—from the observed depth map. This subtraction highlights temporal changes in surface irregularities in the neck region. As the reference map generated from the corrected 3D model may contain missing or sparse data, interpolation is performed using SciPy's griddata function, followed by outlier removal based on the interquartile range (IQR) method. Additionally, a median filter is applied to suppress high-frequency noise.

The resulting laryngeal motion is visualized as a color map representing depth differences. Positive depth differences (indicating motion toward the camera) are shown in progressively darker red, while negative differences (indicating motion away from the camera) are represented in darker blue. From the extracted laryngeal motion map, four regions of interest (ROIs) corresponding to key anatomical landmarks—(A) hyoid bone, (B) cricoid cartilage, (C) sternocleidomastoid muscle, and (D) inferior end of the sternohyoid muscle, as shown in Fig. 5—are analyzed. For each ROI, temporal changes are computed as the average

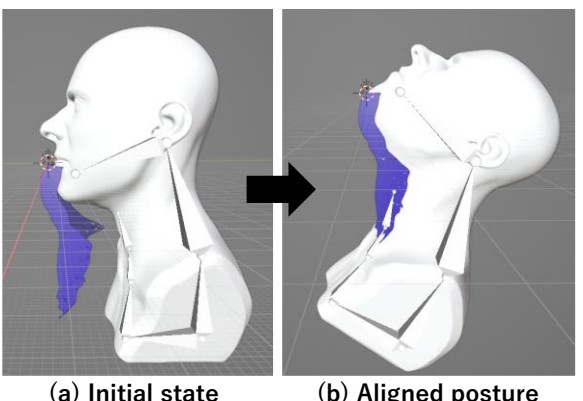

(a) Initial state          (b) Aligned posture

Fig. 4. The corrected 3D head-and-neck model in a static state.

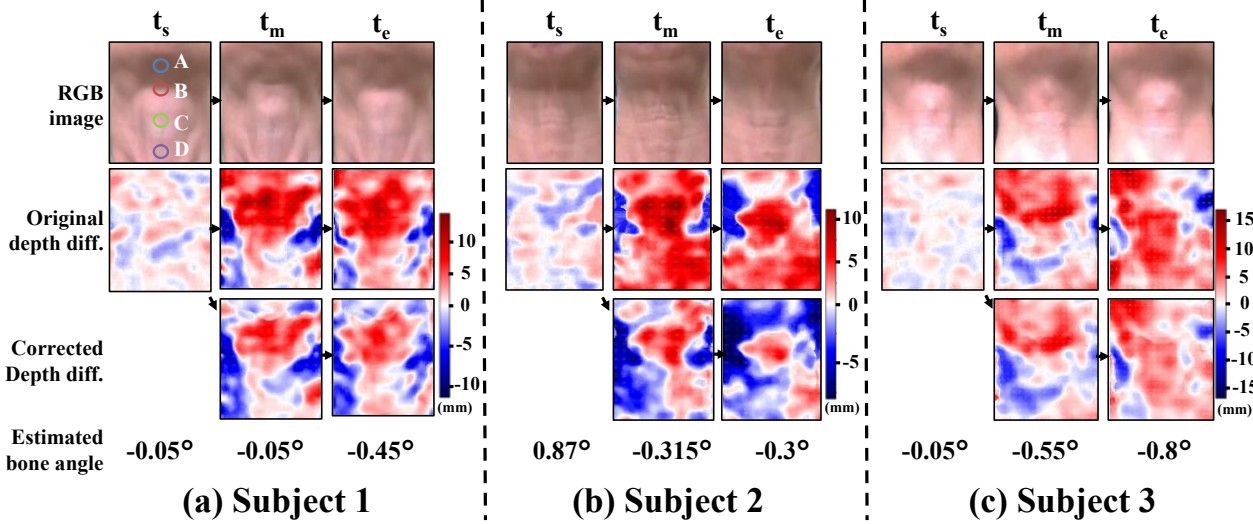

| | $t_s$ | $t_m$ | $t_e$ | | $t_s$ | $t_m$ | $t_e$ | | $t_s$ | $t_m$ | $t_e$ |
|---|---|---|---|---|---|---|---|---|---|---|---|

RGB image — A B C D

Original depth diff.

Corrected Depth diff.

Estimated bone angle

| | $t_s$ | $t_m$ | $t_e$ |
|---|---|---|---|
| **(a) Subject 1** | -0.05° | -0.05° | -0.45° |
| **(b) Subject 2** | 0.87° | -0.315° | -0.3° |
| **(c) Subject 3** | -0.05° | -0.55° | -0.8° |

Fig.5. RGB images of the neck surface, original depth difference, laryngeal movement extracted by the proposed method, and estimated neck bone angle derived by the proposed algorithm at the start of swallowing ($t_s$), maximum elevation ($t_m$), and end of swallowing ($t_e$), respectively.

depth difference within a 5×5 pixel grid centered on manually selected reference points identified through visual inspection.

### E. Experimental Setup and Preprocessing

The water drinking test is commonly used as a screening tool for detecting eating and swallowing disorders [17]. In this test, a small amount (3 mL) of cold water is administered orally using a syringe or cylinder, and the subject is instructed to swallow. The test was conducted with participants seated, and the laryngeal region was recorded from a frontally oblique, lower-angle view, as illustrated in Fig. 3. The distance between the camera and the subject was approximately 35 cm. The image resolution was set to 640 × 360 pixels, with a frame rate of 60 frames per second (fps).

To reduce noise in the depth images, temporal averaging was applied across multiple frames. However, care must be taken to avoid blurring fast laryngeal movements due to excessive averaging. Swallowing consists of four phases: preparatory, oral, pharyngeal, and esophageal phases. Among these, the oral and pharyngeal phases—which involve the transfer of the bolus from the oral cavity to the esophagus—typically occur within approximately 0.6 to 1 second. Considering the trade-off between temporal resolution and noise reduction, we averaged every three consecutive frames of the 60 fps video, resulting in an effective frame rate of 20 fps for subsequent processing.

## IV. EXPERIMENTAL RESULTS

Laryngeal motion during the water drinking test was captured using an RGB-D camera. As shown in the Appendix, we conducted a preliminary experiment in which the proposed method and VFSS were measured simultaneously, and confirmed that similar laryngeal movements could be observed using both approaches. Then, as shown in Table 2, three participants were included in the study: two female (Subjects 1 and 2) and one male (Subject 3). The experiment was conducted with the approval of the ethics review committee of the affiliated institution: No. 202204, 23240(T1). Fig. 5 presents the RGB images of the neck surface at three key time points—start of swallowing ($t_s$), maximum laryngeal elevation ($t_m$), and end of swallowing ($t_e$)—along with the

corresponding original depth differences, laryngeal motion extracted using the proposed method, and estimated neck bone angles (shown in yellow in Fig. 6). These time points were determined by visually inspecting the RGB video frames.

From the color maps obtained at $t_m$ and $t_e$ before correction, it can be observed that the entire neck surface exhibited larger motion artifacts due to overall body movement. In contrast, after applying the proposed correction procedure, localized laryngeal motion was clearly captured. Additionally, Fig. 6 shows the posture of the 3D head and neck model at $t_m$ for each subject. As illustrated, the head posture of the model aligns well with the actual head movement of each participant, confirming that the model accurately reflects subject-specific motion.

Fig. 7 presents the time-series changes in laryngeal motion extracted from regions A to D shown in Fig. 5. A common characteristic observed across all three subjects was that region A exhibited two peaks: one corresponding to the time of maximum laryngeal elevation, and the other coinciding with the peak observed in region B. Additionally, for subjects 2 and 3, the second peak in region A and the peak in region B

TABLE II. PARTICIPANT CHARACTERISTICS

| ID | Sex | Age | Height (cm) | Weight (kg) |
|---|---|---|---|---|
| Subject 1 | Female | 54 | 163 | 43 |
| Subject 2 | Female | 58 | 167 | 65 |
| Subject 3 | Male | 25 | 174 | 66 |

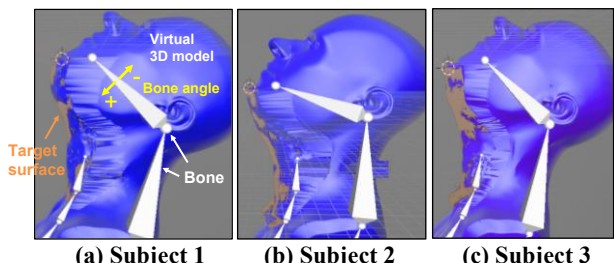

| (a) Subject 1 | (b) Subject 2 | (c) Subject 3 |
|---|---|---|

Fig.6. The posture of the 3D head-and-neck model calculated at the maximum laryngeal elevation.

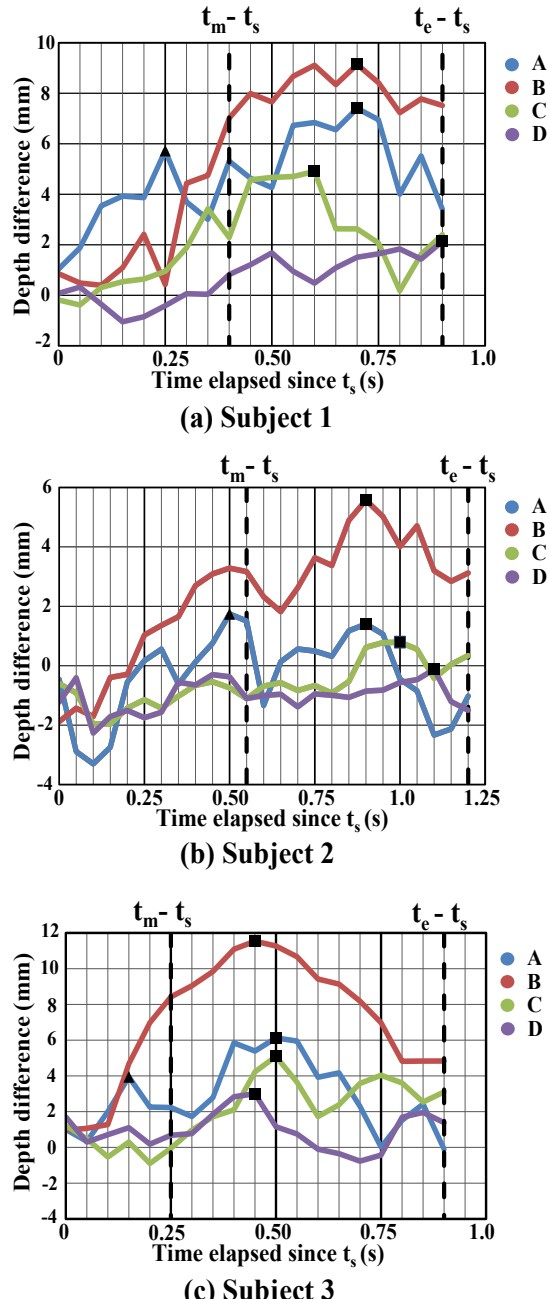

**(a) Subject 1**

**(b) Subject 2**

**(c) Subject 3**

Fig.7. The time series changes in laryngeal movements extracted from areas A to D in Fig. 5.

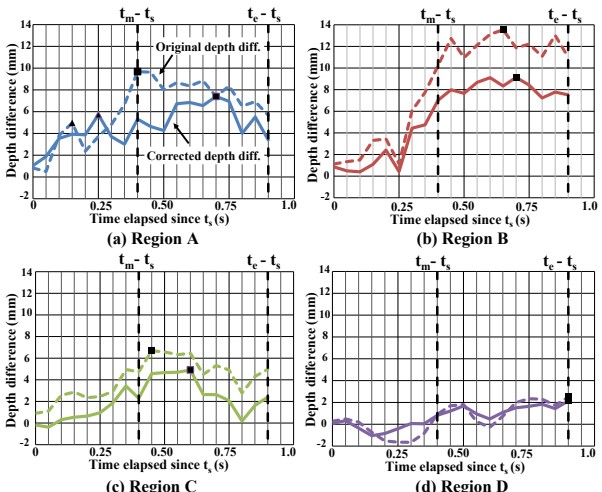

Fig.8. Comparison of laryngeal motion measurements for subject 1 with and without correction using the proposed method; original depth difference (dotted lines) and corrected depth difference (solid lines).

occurred nearly simultaneously, followed by the peak in region C. In contrast, for subject 1, the second peak in region A and the peak in region B appeared later than those in the other subjects.

Fig. 8 illustrates the differences in laryngeal motion measurements for subject 1, comparing cases with and without alignment using the proposed method. Figs. 8(a) through 8(d) correspond to the measurements in regions A through D, respectively. In each graph, the dotted lines represent depth differences computed directly from the original (uncorrected) data, while the solid lines indicate values after applying the proposed correction. Due to a slight forward tilt of the head and neck during the experiment, the

uncorrected measurements show generally larger depth differences across all regions. In region A, the second peak occurs at the time of maximum laryngeal elevation in the uncorrected data. However, after correction, the peak shifts to the timing corresponding to laryngeal descent, suggesting relaxation of the suprahyoid muscle group and return to its resting position. Additionally, since the height of the laryngeal prominence in females is only a few millimeters, the corrected measurements in region C yield more physiologically plausible values near the time of maximum elevation.

Ko, J. Y., Kim et al. [18] reported that the typical order of muscle activation during swallowing is as follows: suprahyoid muscles, sternohyoid muscles, thyrohyoid muscles, and sternothyroid muscles. Based on this, the first peak observed in region A (triangular markers of Fig. 7) was consistently detected across all subjects at approximately 0.15 to 0.5 seconds after the onset of swallowing and is likely related to the activation of the suprahyoid muscles. Furthermore, the peaks in regions B and C at approximately 0.45 to 1.0 seconds are presumed to reflect the activity of the sternohyoid and thyrohyoid muscles, respectively. Based on the above results, it is suggested that it may be possible to confirm the sequence of swallowing actions using a noncontact and unrestrained method.

To evaluate the repeatability of the proposed method, the same subject performed the water drinking test five times. Fig. 9 presents the results of calculating the vertical depth difference between regions A and D at the time of maximum laryngeal elevation $t_m$ in each trial. Fig. 9(a) shows the measurements obtained directly from the original data, while Fig. 9(b) shows those obtained using the proposed method. The five measurement profiles obtained using the proposed method exhibit more similar waveforms, suggesting improved repeatability. The triangular markers in the figures denote the maximum value for each trial. The proposed method yields stable maxima around the laryngeal region (approximately corresponding to region B), which aligns with known anatomical features. The average minimum-maximum distance across the five measurements was 15.0 mm in Fig. 9(a) and 7.7 mm in Fig. 9(b), demonstrating improved measurement stability. Notably, differences were observed in

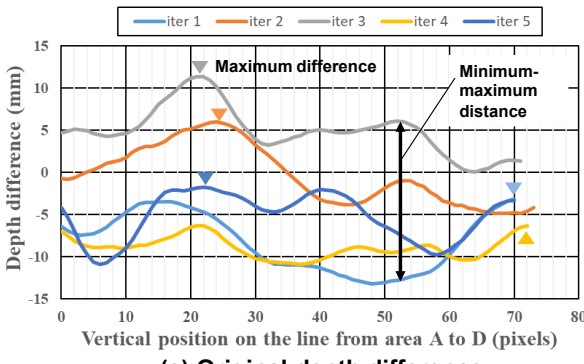

**(a) Original depth difference**

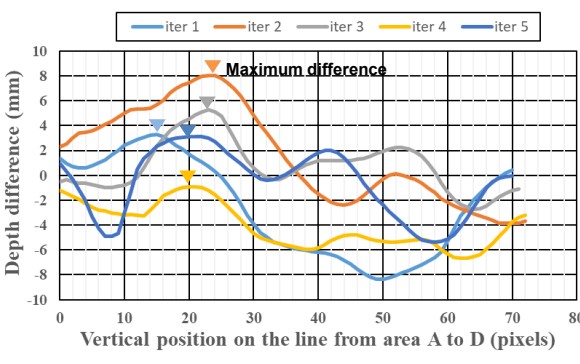

**(b) Corrected depth difference**

Fig.9. Repeatability evaluation based on five water drinking tests; comparing the depth difference along the vertical line from area A to D at maximum laryngeal elevation before and after correction.

region C-D (40-60pixels) between the 1st, 2nd, and 5th trials compared to the 3rd and 4th trials. These discrepancies may reflect actual variations in swallowing conditions across repetitions.

## V. CONCLUSION

We propose a noncontact and unrestrained method for markerless measurement of laryngeal movement using an RGB-D camera. A 3D model of the subject's head is constructed from a depth map acquired in a static state, and body motion is compensated by dynamically deforming the model's posture in virtual space for each frame. Laryngeal movement is quantified by capturing changes in surface irregularities of the neck during a water drinking test.

Preliminary experiments with three subjects demonstrated that the proposed method has the potential to reduce body and neck motion artifacts and to capture localized movements associated with swallowing. The extracted motion patterns were found to be consistent with the activity of muscles involved in swallowing, and common features were observed across participants.

While the results are promising, the small sample size limits the generalizability of the findings. In future work, we aim to increase the number of subjects and apply the method to visualize laryngeal movements during swallowing rehabilitation exercises, such as singing-based training. We will also explore approaches for predicting declines in swallowing function and providing real-time feedback to support personalized intervention.

### ACKNOWLEDGMENT

The authors would like to express their sincere gratitude to Dr. Hidenori Inohara, Dr. Kiyohito Hosokawa, and Dr. Nao Hashida of Graduate School of Medicine, The University of Osaka for their invaluable technical advice and assistance in the acquisition of VFSS data used in this study.

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

As a fundamental experiment, we evaluated the depth measurement accuracy of the RGB-D camera used in our system. Simultaneous image acquisition was performed using both the RGB-D camera and video fluoroscopic swallowing study (VFSS). The skin surface measured from X-ray images was compared with the depth measured by the RGB-D camera. VFSS was conducted from a lateral view, while the RGB-D camera was positioned frontally. The subject was instructed to ingest a contrast agent once and swallow it in two separate portions. To extract the skin surface from the X-ray images, appropriate thresholding and binarization were applied. Depth from a common reference plane were measured in both modalities, and the results for regions A through D are shown in Fig. 10. The mean absolute error (MAE) presented in the figure represents the average difference between the two sets of measurements. In regions B through D, where the skin surface is oriented approximately perpendicular to the camera's line of sight, the measurement error was around 1 mm, indicating that the RGB-D camera can achieve laryngeal motion measurement accuracy comparable to that of VFSS. In contrast, region A is located near the lower jaw, and even

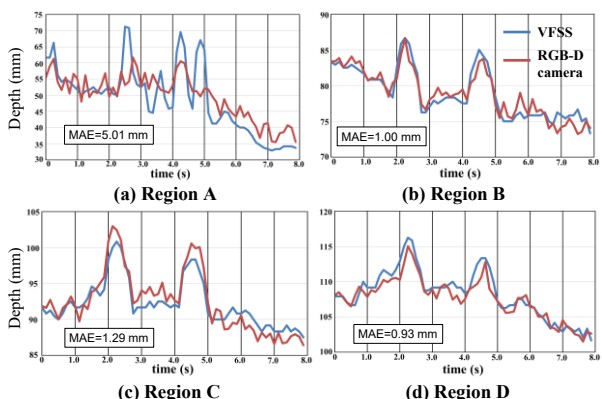

Fig.10. Evaluation of skin surface position measurement accuracy using two measurement methods; VFSS (blue line) and RGB-D camera (red line).

slight misalignment in frontal imaging was found to cause greater discrepancies in distance. Accordingly, in the actual experimental setup, the camera was positioned diagonally below the subject to minimize such errors.