# OpenReview forum: "Markerless Laryngeal Motion Tracking During Swallowing Using an RGB-D Camera with 3D Head-and-Neck Alignment"
_IEEE.org/EMBS/BHI/2025/Conference — BHI 2025_

### Official Review · Reviewer_1TGa · 2025-07-10
**Markerless Laryngeal Motion Tracking During Swallowing Using an RGB-D Camera with 3D Head-and-Neck Alignment - Review**

**Confidence:** 5
**Clarity Of Writing:** great
**Clinical Significance:** excellent
**Methodological Novelty:** excellent
**Overall Rating:** 8

**Experiments And Results:**

excellent

**Questions For The Authors:**

Satisfactory.

**Strengths:**

[1] The solution eliminates the need for physical contact or markers on the skin, avoiding interference with natural swallowing behaviour during examination.
[2] The solution uses a subject-specific 3D head-and-neck model and the Iterative Closest Point (ICP) algorithm to correct for body sway and neck tilt, enabling precise isolation of laryngeal movement with millimetre-level accuracy. This is a promising solution for a real-time, large-scale scenario.
[3] The integration of RGB-D Imaging to capture real-time depth data with virtual anatomical modelling provides a robust framework for capturing complex soft tissue movements. The depth difference calculation and mapping, median filter further make this solution promising for complex muscle activity during swallowing in case of risk.
[4] The paper offers an alternative to Video Fluoroscopic Swallowing Study (VFSS) or endoscopic evaluation of swallowing (FEES), expanding access to dysphagia assessment.

**Summary Of The Paper:**

The paper proposes a markerless laryngeal motion tracking method using an RGB-D camera (Intel RealSense D405) combined with a personalized 3D head-and-neck model to monitor swallowing without physical contact. The method compensates for body sway using the Iterative Closest Point (ICP) algorithm and corrects neck tilt by rotating pseudo-bones in the 3D model to align depth maps across frames. The solution isolates clearer laryngeal movement associated with swallowing from overall body motion by subtracting a corrected static reference depth map from active swallowing frames, and analyzed over four anatomical regions of interest (ROIs): the hyoid bone, cricoid cartilage, sternocleidomastoid muscle, and sternohyoid muscle. In tests with three participants performing a water swallowing task, the method successfully isolated swallowing-related motion and revealed consistent muscle activation patterns aligned with known swallowing physiology. The results suggest the technique offers a radiation-free alternative for real-time monitoring, at-home screening or large-scale settings and rehabilitation for dysphagia and aspiration pneumonia risk.

**Weaknesses:**

[1] The method was tested on three subjects, which may be limited to draw generalizable conclusions about its clinical effectiveness or across population/age variability.
[2] The proposed method was not compared with VFSS (Video Fluoroscopic Swallow Study) or FEES (Endoscopic Evaluation of Swallowing), making it hard to assess its diagnostic accuracy.

---

### Official Review · Reviewer_1W74 · 2025-07-15
**Review of "Markerless Laryngeal Motion Tracking During Swallowing Using an RGB-D Camera with 3D Head-and-Neck Alignment"**

**Confidence:** 4
**Clarity Of Writing:** fair
**Clinical Significance:** good
**Methodological Novelty:** good
**Overall Rating:** 5
**Final Rating:** 6

**Experiments And Results:**

good

**Questions For The Authors:**

•  To what extent is the proposed method generalisable to a broader population, including individuals with varying morphologies, medical conditions, or ethnic backgrounds? This would be important to address given the preliminary nature of the current dataset.
•  What is the current clinical value of the proposed method? If comparative performance data against other techniques or clinician evaluations are not yet available, a qualitative discussion would still be valuable. Ideally, a clear comparison of the method’s performance on key outcomes (relative to existing approaches) would strengthen the study, particularly given its early-stage nature.

**Strengths:**

The paper tackles the important problem of aspiration pneumonia, which affects an ageing global population. The proposed method offers promising translational potential by enabling non-invasive, home-based monitoring. The use of relatively accessible hardware (e.g. RGB-D cameras) increases the applicability of this work across diverse healthcare systems, including in low- and middle-income countries.

**Summary Of The Paper:**

The paper addresses the challenge of non-invasive, home-based monitoring of swallowing function. This is particularly relevant for the management of aspiration pneumonia, a recognised global health concern. Current techniques, such as videofluoroscopy, ultrasound, and wearables, are limited by invasiveness, or impracticality for home use.
To overcome these limitations, the study proposes a markerless method that tracks laryngeal movement during swallowing using an RGB-D camera. This is achieved by using a 3D head–neck model to compensate for head motion in real time, with millimetre-level precision.
Preliminary results from three participants suggest that the proposed method consistently reduces head–neck motion artefacts and allows for reliable extraction of swallowing-related laryngeal motion.

**Weaknesses:**

•  In the abstract, briefly defining “dysphagia” and “aspiration pneumonia” would improve accessibility for non-expert readers.
•  In the abstract, the sentence regarding “monitoring of training effectiveness” is difficult to interpret with no prior context and should be clarified.
•  In the introduction, a typographical error occurs at the end of the sentence “aspiration is becoming an increasingly common global health concern..”; the double period should be corrected.
• In the introduction, several claims in the paragraph beginning with “Because dysphagia often progresses without noticeable symptoms...” require supporting references.
•  In the “Related Work” section, the placement of this section after the methods and aims is unconventional and disrupts the flow. Integrating the content of this section into the introduction (particularly before describing the proposed method) would improve coherence.
•  In the “Related Work” section, several claims would benefit from references, even if drawn from non-peer-reviewed sources (e.g. clinical guidelines from recognised health authorities regarding standard dysphagia care).
•  In the “Related Work” section, Table 1 does not follow IEEE BHI placement requirements and exceeds the recommended width. Table formatting should be revised.
•  Still in the “Related Work” section, the sentence beginning with “Even partial contact between sensors...” is split awkwardly between Figures 1 and 2. Formatting should be adjusted.
•  Figure 1 appears before being cited in the text, which does not conform to IEEE BHI formatting standards.
•  In the methods section, Figures 2 and 4 are similarly placed before being referenced and should be repositioned.
•  In the experimental results section, the “Setup and Preprocessing” subsection would be better placed in the methods section. Additionally, the opening statement would benefit from a supporting reference.
•  The description of participant characteristics is limited. Including age, sex, and potentially anthropometric or demographic information in a structured table would improve transparency and assist in evaluating the generalisability of the findings.
•  In the results section, Figure 5 appears before being referenced. Sentences such as “A—consistently observed across all subjects—is likely associated with the activation of the suprahyoid muscles” are awkwardly phrased and should be revised. Including some numerical values alongside the visual representations would help readers interpret the findings more precisely.
•  In the conclusion, the scientific contributions and limitations of the study are not clearly articulated. Given the small sample size, the claim that the method “effectively suppresses body and neck motion artifacts and enables the extraction of swallowing-related laryngeal movements” may overstate the strength of the findings and should be tempered accordingly.

---

### Official Review · Reviewer_HeH5 · 2025-07-15
**Markerless Laryngeal Motion Tracking During Swallowing Using an RGB-D Camera with 3D Head-and-Neck Alignment**

**Confidence:** 4
**Clarity Of Writing:** great
**Clinical Significance:** great
**Methodological Novelty:** great
**Overall Rating:** 7
**Final Rating:** 7

**Experiments And Results:**

great

**Questions For The Authors:**

- Could the authors clarify whether the algorithm requires baseline data collection prior to swallowing, or whether it is initialized using a generalized static model (e.g., Figure 2) for all subjects? If subject-specific initialization is performed, how long does it take to construct the 3D model of the head at rest?
- The time-series trajectories of laryngeal movements across areas A to D appear to vary considerably between subjects, based on visual inspection. Is this variability expected? Are the authors considering further analysis to explore potential group-wise patterns? Or is the clinical significance tied more to the presence of peaks or other specific temporal features, rather than consistent trajectories?

**Strengths:**

- The introduction and related work sections provide solid background context. I particularly appreciated Table 1, which succinctly summarizes prior literature and highlights the need for this study
- The methodological setup is described in great detail (please see below for the one clarifying question I had)
- The calibration process appears reasonable
- Temporal comparisons of laryngeal movement were well presented in text and figures
- The comparison between aligned and non-aligned cases using the proposed method was compelling, clearly illustrating the benefits of alignment.  I also appreciated the authors’ effort to relate the observed time-series peaks to the typical sequence of muscle activation, which adds meaningful physiological interpretability to the results

**Summary Of The Paper:**

The authors propose a non-contact, non-restraining method for markerless measurement of laryngeal movement using an RGB-D camera. To demonstrate the feasibility of their approach, they conduct a water drinking task (N=4) and analyze laryngeal motion across multiple regions.

**Weaknesses:**

- Including participants with dysphagia (even mild cases) would strengthen the clinical relevance of the results. I wonder if such individuals might employ compensatory neck movements that are not observed in healthy subjects
- The small sample size is a limitation, though understandable given the proof-of-concept nature of the study. As authors have already noted, future work with a larger and more diverse cohort would be valuable.

---

### Official Review · Reviewer_tnkp · 2025-07-16
**Promising Approach for Noninvasive Laryngeal Motion Tracking with Body Movement Correction; Limited by Small Sample Size and Missing Methodological Details**

**Confidence:** 3
**Clarity Of Writing:** fair
**Clinical Significance:** fair
**Methodological Novelty:** good
**Overall Rating:** 6
**Final Rating:** 7

**Experiments And Results:**

fair

**Questions For The Authors:**

- It is unclear whether the estimated neck bone angle in Figure 5 is manually annotated or algorithmically derived—this should be explicitly stated.
- Figure 4  Figure 3: Does Fig. 3 correspond to 'body sway alignment' from Fig. 4?
- It’s also unclear what is meant by "incremental rotation"(in 'Neck Tilt Alignment')—what is increment?
- Is there a way to quantify group statistics or normalized curve comparison (as done in previous literature, e.g., https://doi.org/10.1038/s41598-018-23486-0 ). Some kind of group pattern estimation would strengthen the argument.
- The explanation of the alignment procedure is vague: “bone rotation minimizes the distance between the lowest z-coordinates” is hard to interpret; please illustrate/clarify. From Fig 3, it does not seem that the lowest points of the neck mesh reach low enough to the lowest point of the 3D reference model.
- Is the reference 3D model adopted from elsewhere? Please specify.
- Consider discussing: small sample size, anda  human in the loop needed

**Strengths:**

The manuscript is well-organized, with a clear motivation and well-defined problem statement.
It proposes a novel noninvasive method for estimating laryngeal motion using a depth camera and 3D modeling.
The use of a 3D face model to correct for body sway and neck tilt adds value to the motion tracking process.
The manuscript clearly visualize laryngeal motion data and demonstrate the effect of motion correction, highlighting the importance of alignment.

**Summary Of The Paper:**

This manuscript proposes a novel noninvasive approach to estimate laryngeal motion using depth cameras. The method involves fitting a 3D face model to subject-specific data and tracking laryngeal movement during swallowing. To account for head and neck motion, the approach includes a human-in-the-loop step for alignment and correction. The method was applied and validated on data from three subjects.

**Weaknesses:**

- The approach may be difficult to scale, as it appears to require human intervention or manual alignment, which could limit real world adoption.
- The sample size (n=3) is very small. This limitation could be openly acknowledged and discussed.
- Besides showing individual curves, some kind of group pattern estimation would strengthen the argument.
(as done in previous literature, e.g., https://doi.org/10.1038/s41598-018-23486-0 ).
- The source and creation process of the 3D reference model is not explained. If it is an external model, the manuscript should clarify how it was obtained or constructed.
- Experimental and demographic details are missing: Camera specs, participant characteristics (e.g. age, sex), normal swallowing or not.
The manuscript lacks comparisons with prior work, at least in the discussion section.
- Additional clarifications would improve reproducibility and strengthen the argument, as noted in 'Questions for authors'

---

### Official Review · Reviewer_rnur · 2025-07-17
**Novel non-contact approach to laryngeal motion tracking with limited validation**

**Confidence:** 4
**Clarity Of Writing:** good
**Clinical Significance:** good
**Methodological Novelty:** good
**Overall Rating:** 6
**Final Rating:** 7

**Experiments And Results:**

fair

**Questions For The Authors:**

1. How consistent are the measurements across multiple sessions or repeated trials on the same subject? This would help assess reliability.
2. How does the model handle occlusions or poor lighting conditions, which are common in non-clinical environments?
3. Can the method be extended to track more dynamic or complex exercises, such as singing or rehabilitation movements?

**Strengths:**

1. The method is fully markerless and contact-free, reducing subject burden and increasing usability in home settings.
2. Combines computer vision with physiological modeling for a practical healthcare application.
3. The use of a personalized 3D model and motion compensation improves robustness compared to earlier RGB-D approaches.

**Summary Of The Paper:**

The paper proposes a non-contact method for markerless tracking of laryngeal motion during swallowing. It builds a subject-specific three-dimensional head and neck model from static depth images and then aligns this model with sequential depth maps to remove body sway and vertical neck movement. Laryngeal motion is extracted by subtracting the aligned reference map from observed frames. The method is tested on a water-drinking test using an RGB and depth camera at sixty frames per second with three participants. Results demonstrate artifact reduction and reveal consistent depth displacement patterns in regions corresponding to anatomical landmarks.

**Weaknesses:**

1. The evaluation is limited to only three subjects, with minimal demographic variation or control. This restricts generalizability.
2. No discussion of inter-session or intra-subject variability, which is important for real-world deployment.
3. The paper does not compare the method’s accuracy directly against a clinical standard like VFSS.